

# A multimodal prediction model for suicidal attempter in major depressive disorder

Qiaojun Li[1,*] and Kun Liao[2,*]

[1] College of Information Engineering, Tianjin University of Commerce, Tianjin, China
[2] College of Sciences, Tianjin University of Commerce, Tianjin, China
[*] These authors contributed equally to this work.

## ABSTRACT

**Background**. Suicidal attempts in patients with major depressive disorder (MDD) have become an important challenge in global mental health affairs. To correctly distinguish MDD patients with and without suicidal attempts, a multimodal prediction model was developed in this study using multimodality data, including demographic, depressive symptoms, and brain structural imaging data. This model will be very helpful in the early intervention of MDD patients with suicidal attempts.

**Methods**. Two feature selection methods, support vector machine-recursive feature elimination (SVM-RFE) and random forest (RF) algorithms, were merged for feature selection in 208 MDD patients. SVM was then used as a classification model to distinguish MDD patients with suicidal attempts or not.

**Results**. The multimodal predictive model was found to correctly distinguish MDD patients with and without suicidal attempts using integrated features derived from SVM-RFE and RF, with a balanced accuracy of 77.78%, sensitivity of 83.33%, specificity of 70.37%, positive predictive value of 78.95%, and negative predictive value of 76.00%. The strategy of merging the features from two selection methods outperformed traditional methods in the prediction of suicidal attempts in MDD patients, with hippocampal volume, cerebellar vermis volume, and supracalcarine volume being the top three features in the prediction model.

**Conclusions**. This study not only developed a new multimodal prediction model but also found three important brain structural phenotypes for the prediction of suicidal attempters in MDD patients. This prediction model is a powerful tool for early intervention in MDD patients, which offers neuroimaging biomarker targets for treatment in MDD patients with suicidal attempts.

## INTRODUCTION

Major depressive disorder (MDD) is a disabling mental disorder regulated by both genomics and environment (*Otte et al., 2016*), currently affecting more than 320 million people in the world (*Zhuo et al., 2019*) and causing a serious burden on society (*Whiteford et al., 2013*).

Corresponding author
Qiaojun Li, liqiaojun@tjcu.edu.cn

Suicide refers to the behavior of ending one own's life (*Sveticic & De Leo, 2012*), which is a serious consequence of global mental health and contributes to one million deaths each year (*Brundin, Bryleva & Rajamani, 2017*). Nowadays, 90% of people with suicide attempts suffer from one or more mental disorders, and those with MDD account for 59–87% of all suicidal attempts (*Rihmer & Kiss, 2002*). Therefore, it is extremely important to predict the suicide attempts of MDD patients and explore underlying neural mechanisms to alleviate the harm of suicide attempts.

Increasing evidence has demonstrated significant demographic and clinical differences between MDD patients with and without suicidal attempts. For example, suicidal MDD patients have more comorbid alcohol dependence (*Schick et al., 2023*) and experience frequent nightmares (*Song et al., 2022*). They were also found to have an earlier onset age of MDD (*Claassen et al., 2007*), lower education (*Wang et al., 2022*) and social support (*Hu et al., 2023a*), more vulnerability to external control (*Wiebenga et al., 2021*) and more often exposed to childhood trauma (*Souza et al., 2016*). With the development of magnetic resonance imaging, more and more studies are showing significant alterations in brain structure in MDD patients who attempt suicide. For example, suicidal attempters demonstrated larger bilateral hippocampal fissures (*Zhang et al., 2021*) but smaller hippocampal volumes (*Colle et al., 2015*). MDD patients with suicide attempts showed larger surface area in the left posterior central region and lateral occipital region (*Kang et al., 2020*), larger prefrontal cortical and insula volume (*Rizk et al., 2019*), and smaller left angular gyrus and right cerebellum volume (*Lee et al., 2016*).

Although the behavioral and neuroimaging differences among MDD patients with and without suicide attempts were inconclusive (*Dwivedi, 2012*), they still offered some clues for the prediction of suicide attempts in MDD patients. Increasing machine learning (ML) methods have been conducted for the prediction of suicide attempts in MDD patients using behavioral and neuroimaging data, in which recursive feature elimination (RFE) and its variation have been widely adopted (*Bhadra & Kumar, 2023*). RFE has shown excellent performance in feature selection from high-dimensional data. For example, history of suicide attempts, religion, ethnicity, suicidal ideation, and severity of clinical depression are sensitive in predicting suicide attempters in MDD using RFE (*Nordin et al., 2021*). Support vector machine-recursive feature elimination (SVM-RFE) integrates the advantages of SVM and RFE by filtering and ranking features (*Bao et al., 2023*). SVM-RFE has been established to be a useful feature elimination method in predicting suicide attempt risk among MDD patients (*Hong et al., 2021*), with orbitofrontal, cingulate, fusiform, and temporal pole thickness and volume as important selected features. In addition, Random Forest (RF) is a traditional feature selection algorithm that ranks feature importance by calculating contribution weight into a decision tree (*Gündoğdu, 2023*), which can stratify suicide risk among MDD patients, with pain avoidance and right thalamus volume as key features selected from behavioral and neuroimaging data (*Hao et al., 2023*). These studies show that the isolated ML approach may help to provide an individual-level prediction of suicidal risks in MDD patients (*Chen et al., 2023*), however, integrated ML multimodal in predicting suicide attempt risk among MDD patients using high-dimensional behavioral and neuroimaging data have been less investigated.
In this study, to accurately predict suicide attempt risk among MDD patients, we developed an ML suicidal multimodal prediction model in MDD patients in the UK Biobank dataset using high-dimensional behavioral and neuroimaging data. We aim to optimize accuracy in predicting suicide attempt risk among MDD patients using an integration of SVM-RFE and RF models. In addition, the surviving selected neuroimaging features will strengthen the understanding of neural mechanisms in suicide attempts in MDD patients. Our suicidal multimodal prediction model can not only be used for early prediction of suicide attempts in MDD patients but also offers neuroimaging biomarker targets for treatment in MDD patients.

## MATERIAL AND METHODS

### Study populations

The participants used in this study were from the UK biobank (UKBB) (http://www.ukbiobank.ac.uk), which is a population-based cohort including over 500,000 participants recruited in the United Kingdom between 2006 and 2010 (*Sudlow et al., 2015*). Among the 502,616 participants, exclusively adults, with age ranges from 40 to 77, the mean age at baseline was 59.46 years (standard deviation 8.12), 54.41% were men and 81.51% were of White ethnicity. UKBB is used to improve the prevention, diagnosis, and treatment of various diseases. The baseline assessment includes lifestyle, environment, medical history, genomics, physical measures, and other relevant data. Informed consent was obtained from all UKBB participants.

This study was subject to UKBB ethical approval granted by the National Information Governance Board for Health and Social Care and the NHS North West Multicenter Research Ethics Committee. All participants gave informed consent by electronic signature at baseline. Data collected at baseline were used in this study. This study was conducted using the UK Biobank Resource under Application Number 75556.

### Definition of major depressive disorders

The definition of MDD was based on major depression status in the UKBB(field ID: 20126). The definition of MDD in this field is generated based on the hypothetical categories of MDD summarized by *Smith et al. (2013)*, which are classified into single-episode major depression, moderate recurrent major depression, and severe recurrent major depression. For participants with missing data, we excluded them from the study. Finally, there were a total of 31,829 MDD patients included in this study.

### Definition of suicidal attempts

The definition of suicidal attempts was based on "Ever attempted suicide" (field ID: 20483). Each participant was asked to answer a question about "Have you harmed yourself with the intention to end your life?". In the latest data released by UKBB, excluding those participants with the answer "Prefer not to answer" ($n = 219$), there were a total of 6,642 participants answering this question. Finally, there were a total of 1132 MDD patients with completed the suicidal attempts questionnaire. Among 1,132 MDD patients, there were a total of 653 participants defined as MDD patients with suicide attempts (MDD-SA,

$n = 653$), and a total of 479 participants defined as MDD patients without suicide attempts (MDD-nSA, $n = 479$).

## Features characteristics

A total of 179 features were included in the prediction analysis, including demographic features ($n = 11$), depressive symptoms features ($n = 15$), and brain structural phenotypes features( $n = 153$). More information about these features is described in Table S1.

*Demographic features.* There were eleven demographic features, including gender, age at the time of attending the assessment center, current employment status, age at completion of full-time education, moral background, age at the time of the first episode of depression, age at the time of the last episode of depression, sleep duration, smoking status, variation in diet and alcohol intake frequency.

*Depressive symptoms feature.* There were fifteen depressive symptoms features, for example, "numbers of depression episodes", "even had prolonged feelings of sadness or depression", "feeling of tiredness during a worst episode of depression", and so on. More information about these features is described in Table S1.

*Brain structural phenotypes.* In this study, neuroimaging data were scanned at the UK Biobank Imaging Centre by a standard Siemens Skyra 3T scanner running VD13A SP4 and a standard Siemens 32-channel radiofrequency receiver head coil (*Miller et al., 2016*). More information is shown at https://biobank.ctsu.ox.ac.uk/crystal/crystal/docs/brain_mri.pdf. The FAST and FIRST grey matter segmentation are used to generate a further 153 regional brain structural phenotypes, by summing the grey matter partial volume estimates within 153 ROIs: 139 regional grey matter volumes (GMV) and 14 subcortical volumes (Category ID: 1101 and 1102). These ROIs are established in MNI152 space by merging several different atlases: the Harvard-Oxford cortical and subcortical atlases (https://fsl.fmrib.ox.ac.uk/fsl/fslwiki/Atlases) and the Diedrichsen cerebellar atlas (http://www.diedrichsenlab.org/imaging/propatlas.htm). More details are available at https://biobank.ctsu.ox.ac.uk/crystal/crystal/docs/brain_mri.pdf. A total of 42,789 participants' neuroimaging data were used for the present study. Due to the small amount of data, we excluded fields with missing rates exceeding 10% from the study. For the remaining categorical variables, the missing values were filled in using the mode value. For the remaining continuous variables, the missing values were filled in using mean and then standardized to z-score for further analyses.

Finally, 208 MDD patients (MDD-SA, $n = 119$; MDD-nSA, $n = 89$) with complete 11 demographic features, 15 depressive symptoms features, and 153 brain structural phenotypes features were included in the further analysis. A flowchart for describing the sample size in each analysis is shown in Fig. S1.

## Statistical analysis

In this study, an ML multimodal model was built for the prediction of suicide attempters in MDD patients by integrating SVM-RFE and RF algorithms in 208 MDD patients. The process includes feature selection, feature integration, and prediction model estimation. The working flowchart in this study is described in Fig. 1 and details are introduced below.

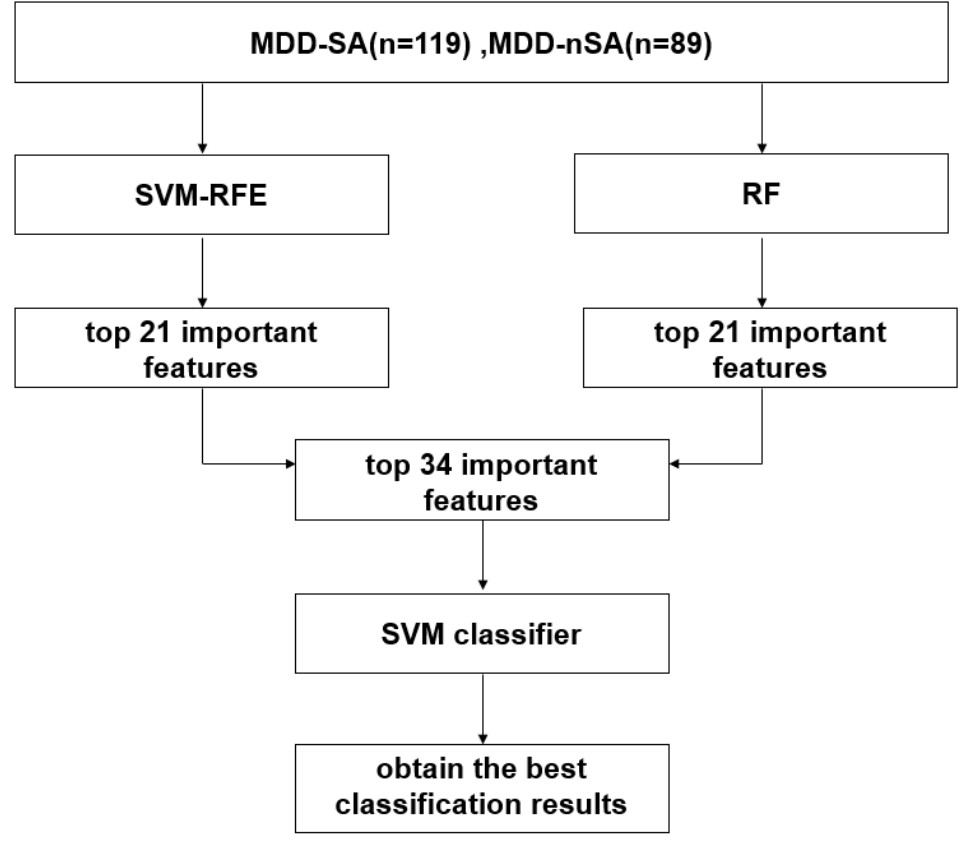

**Figure 1** **Working flowchart in the present study.** Abbreviation: SVM, support vector machine; SVM-RFE, support vector machine-recursive feature elimination; RF, random forest; MDD-SA, MDD patients with suicidal attempts; MDD-nSA, MDD patients without suicidal attempts.

## Feature selection

Feature selection is an important step in the classification prediction model. In this study, two classical feature selection methods, namely SVM-RFE and RF, were used to explore important features in the classification of MDD-SA and MDD-nSA. This study implemented ML feature selection using the scikit-learn package with Python 3.7 (http://scikit-learn.org/). SVM-RFE algorithm is a feature selection algorithm for recursive feature elimination based on the principle of maximum interval of SVM, in which the three most important parameters are kernel function, kernel function coefficient gamma, and penalty coefficient C. In this study, we set the kernel function to be "linear kernel", the kernel function coefficient to be "auto", and the penalty coefficient to be 1. RF is an algorithm for feature selection by calculating the contribution of each feature on each tree in a random forest, the two most important parameters are the number of decision trees and the classification criteria of the tree nodes criterion, which we set to 50000 and "Gini", respectively. More information about the algorithm parameters used in the experiments is shown in Table S2.

The specific implementation process of the SVM-RFE method was as follows: (1) All 179 features were introduced into SVM-RFE to train the classifier, in which the less important features were iteratively eliminated according to the weighted vector of SVM with linear kernel, and the features were rearranged in descending order according to their importance; (2) Repeated training and testing split design was conducted using leave-one-out cross-validation (LOOCV). Specifically, in each experiment, one dataset was used for model test, and all other data was used for model training. The feature importance ranking was recorded according to all the iterative results. (3) Finally, the predictive features that survived from Subset-SVM-RFE were adopted for the following analysis.

The implementation of the RF method was as follows: (1) All 179 features were introduced into an RF classifier to build 50,000 subtrees, using Gini Impurity as an evaluation criterion for dividing the subtrees (*Menze et al., 2009*); (2) The features were ranked by importance and the top features were selected as subset-RF for subsequent analysis. The numbers of top features in Subset-RF were consistent with the numbers of features in Subset-SVM-RFE.

## Feature integration

The features in Subset-SVM-RFE and Subset-RF were then integrated for further analysis. The common features in Subset-SVM-RFE and Subset-RF were treated as Features-Subset and other features were added to the Features-Subset one by one based on their importance for model training. The dataset for model training was stratified into train and test datasets with a ratio of 7:3, ensuring the category proportion. A linear kernel was used to fit the SVM model, the rest of the parameters was based on the default parameters of the SVM model. More information about the algorithm parameters used in the experiments is shown in Table S2.

## Prediction model estimation

Three indices in prediction model estimation, including accuracy, sensitivity, and specificity, were used to estimate the prediction model. Additionally, two important statistics, positive predictive value (PPV) and negative predictive value (NPV) were used to measure the optimized clinical relevance of a test. The receiver operating characteristic (ROC) curve for this optimal model was also demonstrated. To validate the selected features that survived from the integrated prediction model, a two-sample *T-test* of the predictors in the resulting optimal classifier was performed to compare the differences in features among MDD-SA and MDD-nSA participants.

# RESULTS

## Demographic

A total of 208 MDD patients were finally included in this study, their demographic and clinical characteristics are shown in Table 1. There were no statistical differences between MDD-SA ($n = 119$) and MDD-nSA ($n = 89$) in terms of sex, current employment status, sleep duration, smoking status, dietary changes, and frequency of alcohol consumption (Table 1). The age at the time of assessment in the MDD-SA group ($53.09 \pm 7.51$ years) was significantly higher than those in the MDD-nSA group ($50.54 \pm 7.07$ years) ($P = 0.015$).

**Table 1 Demographic information of MDD patients.**

| | MDD-SA ($n = 119$) | MDD-nSA ($n = 89$) | $P$-value |
|---|---|---|---|
| **Sex (male/female)** | 31/88 | 20/69 | |
| **Age (years)** | 53.09 ± 7.514 | 50.54 ± 7.068 | *0.02*[a] |
| **Current employment status** | | | 0.92[b] |
| In paid employment or self-employed | 91 (76.47%) | 66 (74.16%) | |
| Retired | 17 (14.29%) | 14 (15.73%) | |
| Looking after home and/or family | 2 (1.68%) | 2 (2.25%) | |
| Unable to work because of sickness or disability | 7 (5.88%) | 4 (4.49%) | |
| Unemployed | 2 (1.68%) | 2 (2.25%) | |
| Full or part-time student | 0 | 1 (1.12%) | |
| **Sleep duration** | 7.08 ± 1.277 | 7.15 ± 1.029 | 0.55[a] |
| **Smoking status** | | | 0.83[b] |
| Never | 58 (48.74%) | 47 (52.8%) | |
| Previous | 50 (42.02%) | 35 (39.33%) | |
| Current | 11 (9.24%) | 7 (7.87%) | |
| **Variation in diet** | | | 0.68[b] |
| Never/rarely | 36 (30.25%) | 22 (24.72%) | |
| Sometimes | 72 (60.5%) | 58 (65.17%) | |
| Often | 11 (9.25%) | 9 (10.11%) | |
| **Alcohol intake frequency** | | | 0.73[b] |
| Daily or almost daily | 18 (15.13%) | 14 (15.73%) | |
| Three or four times a week | 26 (21.85%) | 25 (28.09%) | |
| Once or twice a week | 36 (30.25%) | 22 (24.72%) | |
| One to three times a month | 18 (15.13%) | 11 (12.36%) | |
| Special occasions only | 10 (8.40%) | 11 (12.36%) | |
| Never | 11 (9.24) | 6 (6.74%) | |

**Notes.**

MDD-SA, MDD patients with suicidal attempts; MDD-nSA, MDD patients without suicidal attempts.

Statistically significant difference ($P < 0.05$) of each item among MDD-SA and MDD-nSA were shown in bold and iliac.

[a]$P$-value in Mann-Whitney U test.

[b]$P$-value in chi-square test.

## Feature selection and integration

In the feature selection using SVM-RFE, the Subset-SVM-RFE selected a total of 21 predictive features. Correspondingly, the Subset-RF included the top 21 predictive features from the RF algorithm (Table S3). The SVM-RFE and RF algorithms were integrated to explore useful prediction features for the classification of MDD-SA and MDD-nSA. We found that there were eight common features between Subset-SVM-RFE and Subset-RF. The other 34 features from Subset-SVM-RFE and Subset-RF were added to the predictive model interactively to estimate the predictive performance. More information about these features is shown in Table S3.

## Predictive model performance

In this study, three model prediction indices, including accuracy, sensitivity, and specificity, were used to estimate the predictive model. The process of adding features one by one

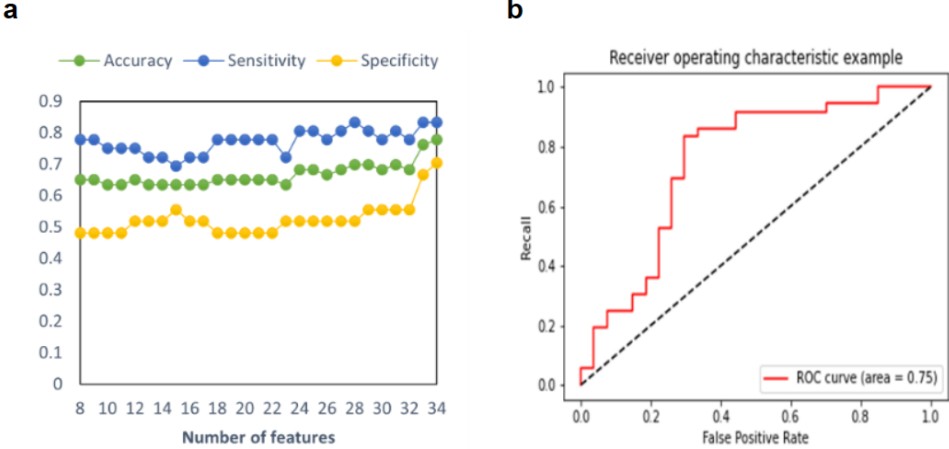

**Figure 2  Multimodal predictive model performance for MDD-SA.** (A) Accuracy, sensitivity, and specificity in the prediction model for MDD-SA; (B) the ROC curves for the prediction model. Abbreviation: ROC: receiver operating characteristic. Abbreviation: MDD-SA, MDD patients with suicidal attempts; ROC, Receiver Operating Characteristic.

**Table 2  Comparison of the classification performance of different feature selection methods.**

|  | ACC | TPR | TNR | PPV | NPV | AUC |
|---|---|---|---|---|---|---|
| SVM-RFE | 76.19% | 80.56% | 70.37% | 78.38% | 73.08% | 0.75 |
| RF | 65.08% | 77.78% | 48.15% | 66.67% | 61.90% | 0.62 |
| SVM-RFE and RF | 77.78% | 83.33% | 70.37% | 78.95% | 76.00% | 0.75 |

Notes.

ACC, accuracy; AUC, area under the curve; NPV, negative predictive value; PPV, positive predictive value; RF, random forest; SVM- RFE, support vector machine- recursive feature elimination; TNR, true negative rate; TPR, true positive rate.

to achieve the best classification effect is shown in Fig. 2. As shown in Table 2, in the prediction model of a single feature selection method, RF demonstrated an accuracy of 65.08% of predictive performance, and SVM-RFE demonstrated an accuracy of 76.19%. After iteratively integrating features from Sub-SVM-RFE and Sub-RF, we found that the predictive model with 34 features demonstrated the best performance, with a balanced accuracy of 77.78%, a sensitivity of 83.33%, and a specificity of 70.37%. Additionally, two important statistics, PPV and NPV were used to measure the best clinical relevance of a test. In this study, the PPV of MDD-SA was 78.95%, and the NPV value was 76.00%. The ROC curve with AUC =0.75 for this optimal model is shown in Fig. 2. For the prediction model of integrated feature selection methods, its prediction accuracy is nearly 13% higher than that of feature selection using RF, and 1.59% higher than that of the model using SVM-RFE for feature selection. This may be because the SVM-RFE algorithm calculates feature scores based on SVM, and the selected features can fit the model well when reintroduced into the SVM predictor for prediction, resulting in roughly equal accuracy between the two algorithms.

This study additionally conducted a post-hoc analysis on the finally used features in the model, and significant differences were found in 9 features from the above 34 features

**Table 3** Significant differences of nine predictive features from multimodal predictive model for MDD-SA.

| Feature name | Statistics | P-value |
|---|---|---|
| **Brain structural phenotypes features** | | |
| Right hippocampal volume | 2.02 | *0.045* |
| Left hippocampal volume | 2.09 | *0.038* |
| Left supracalcarine cortex volume | −2.56 | *0.011* |
| Left precuneous cortex volume | −2.17 | *0.031* |
| Right thalamus volume | −2.22 | *0.028* |
| Right precuneous cortex volume | −1.99 | *0.048* |
| **Demographics features** | | |
| Age when attended assessment centre | −2.44 | *0.015* |
| **Depressive symptom features** | | |
| Number of depression episodes | −2.43 | *0.015* |
| Feelings of worthlessness during worst period of depression | 9.38 | *0.002* |

**Notes.**
MDD-SA, MDD patients with suicidal attempts.
*P* value with statistically significant difference were shown in bold and italic.

between MDD-SA and MDD-nSA. The brain structural phenotypes feature included bilatary precuneus cortex volume, bilateral hippocampal volume, left supracalcarine cortex volume, and right thalamus volume. The demographic and behavioral features included "age when attending assessment centre", "number of depression episodes", and "feelings of worthlessness during worst period of depression" (Table 3).

## DISCUSSION

In this study, we developed an ML suicidal multimodal prediction model in MDD patients in the UK Biobank dataset using high-dimensional behavioral and neuroimaging data. Using the integration of SVM-RFE and RF models, accuracy in predicting MDD-SA could be optimized compared to any of the single feature selection methods. We also selected the neuroimaging features, which will strengthen our understanding of neural mechanisms in MDD-SA patients. Our suicidal multimodal prediction model can not only be used for early prediction in MDD-SA patients but also offers neuroimaging biomarker targets for treatment in MDD patients.

Exploring the prediction possibility in MDD-SA patients is a very important research topic and has aroused some applicable findings (*Hu et al., 2023b*; *Zheng et al., 2022*). In this study, our algorithm showed a prediction accuracy of 77.78%, a sensitivity of 83.33%, and a specificity of 70.37%, which outperformed than any one of the selection methods. It can be attributed to two factors, the used features for prediction and prediction methods. Shuqiong Zheng et al. only adopted cognition features and Jinlong Hu et al. only used structural MRI features, which hindered the exploitation of comprehensive information from different modalities. Here, we took advantage of multimodal feature fusion by integrating behavioral, demographic, and brain structural phenotype data.

The feature selection method is also very vital for prediction performance. Compared to the previous studies using only one method (*Hong et al., 2021*), we proposed a strategy of combining two general feature selection methods, SVM-RFE and RF algorithms. SVM-RFE has been widely used to do feature selection in neuroimaging (*Guyon et al., 2002*), in which the importance of each original feature is directly related to its weight coefficient, thus allowing simple identification of the most discriminative features in original data. RF algorithm is also popular in high-dimensional data feature selection, in which the input features are sorted in descending order according to their importance. In this study, we merged these two feature selection methods to include as many features as possible.

In this study, a total of 34 predictive features were finally adopted for our prediction model, to explain the classification variables that distinguish between MDD-SA and MDD-nSA, we conducted two-sample $t$-tests on all 34 predictive features to test for inter-group differences. Left hippocampal volume, cerebellar volume, and left supracalcarine cortex volume were found to be the most important three predictive features. The hippocampus is a very important brain structure, and its volume was found to be associated with episodic memory (*Tulving & Markowitsch, 1998*), emotion regulation (*Barch et al., 2019*), and attention (*Kim et al., 2021*). Consistent with previous studies (*Colle et al., 2015*), hippocampal volume was found to play a very important role in the prediction of MDD-SA, presenting statistical differences between MDD-SA and MDD-nSA patients. The cerebellum plays a key role in sensory motor and vestibular control, as well as in emotional and autonomic functions (*Schmahmann, 2019*). Its volume is related to working memory performance (*Ding et al., 2012*), high-order cognition (*Buckner, 2013*), sensorimotor control (*Manto & Ben Taib, 2013*), and here we found for the first time that cerebellar volume can be a significant predictor in MDD-SA. The supracalcarine cortex volume was found to associate with cognitive function (*Dichter et al., 2009*), and this is also the first time to be found that it can be referred to as an important predictive feature in MDD-SA patients.

Our research has advanced efforts to predict MDD-SA patients; however, more work still needs to be done for the application into clinical assessment for identifying MDD-SA patients. A limitation of the present study is our participants range from 49 to 77 in the UK Biobank dataset, the generalization into other age populations still needs to be explored. More external validation should be tested based on our study. Exploration of involving other modality features in the prediction model for MDD-SA could be tested in the future.

### Funding

This work received support from the Natural Science Foundation of Tianjin (22JCQNJC01450 for Qiaojun Li) and the Tianjin Postgraduate Research Innovation Project (2022SKYZ133 for Kun Liao). The funders had no role in study design, data collection and analysis, decision to publish, or preparation of the manuscript.

## Grant Disclosures

The following grant information was disclosed by the authors:
Natural Science Foundation of Tianjin: 22JCQNJC01450.
Tianjin Postgraduate Research Innovation Project: 2022SKYZ133.

## Competing Interests

The authors declare there are no competing interests.

## Author Contributions

- Qiaojun Li conceived and designed the experiments, authored or reviewed drafts of the article, and approved the final draft.
- Kun Liao conceived and designed the experiments, performed the experiments, analyzed the data, prepared figures and/or tables, and approved the final draft.

## Data Availability

The original data and code are available in the Supplementary Files.

## Supplemental Information

Supplemental information for this article can be found online at http://dx.doi.org/10.7717/peerj.16362#supplemental-information.

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
