# Peer review of "A multimodal prediction model for suicidal attempter in major depressive disorder"

_PeerJ, doi:10.7717/peerj.16362_

## Round 0.1 · original submission · Major Revisions

Please clarify multiple methodological issues raised by the reviewers.

**Language Note:** The review process has identified that the English language must be improved. PeerJ can provide language editing services - please contact us at copyediting@peerj.com for pricing (be sure to provide your manuscript number and title). Alternatively, you should make your own arrangements to improve the language quality and provide details in your response letter. – PeerJ Staff

Reviewer 1 ·

Basic reporting

The aim of this manuscript was to develop a machine learning model for predicting suicide attempts in patients with major depressive disorder (MDD). It is a good attempt to compare two feature selection methods and find the best feature for predicting suicide attempts. Overall, the manuscript is relevant and significant to the field of psychiatry as it presents the selected features using feature selection methods that achieve moderate performance in terms of accuracy, sensitivity and specificity. However, there are some critical issues that the authors should consider in improving the current manuscript.

1) In the Abstract, the author has not described the problem and objective of the study, which is important for readers to know what the manuscript is about.
2) Also, in the Abstract, the conclusion statement is confused. Does the combination of two feature selection methods outperform traditional methods? Is it a combination of two feature selection methods or a combination of features from two selection methods?
3) The keywords do not include feature selection. It is recommended to include feature selection as one of the keywords.
4) English spelling and grammar also need to be checked carefully. For example, it should be ‘prediction model’ and not ‘predication model’.
5) The original code provided by the authors can be easily explored, but the raw data given is difficult to read. Please provide metadata identifiers that are more descriptive and informative.

Experimental design

1) In the Introduction, the problem of the study is not clearly stated, and the contribution of the study compared to other studies is lacking. The objective is general and need to be specific to the study that has been conducted. Overall, the introduction section needs to clearly explore and explain the existing studies on suicidal attempt prediction.
2) In the Materials and Methods, the study uses the UK Biobank Dataset, which focuses on MRI image data. In the ‘definition of suicidal attempt’, there are 6643 people in the data field. However, only 208 peoples (subjects) were used in this study. How do the authors derive from 6643 subjects to 208 subjects?
3) How many of the 208 subjects are suicide attempters and non-suicide attempters in the dataset?
4) Was any data pre-processing done before training the model, e.g.: data cleaning, check missing value?
5) The results mentioned is good as shown in Table 2, which compares the features selected by SVM-RFE and RF. However, are the features of the clinical demographic listed in Table 1 also included as input for training the model?

Validity of the findings

The discussion is clear and justified with existing work on the feature selected by the two selection methods. However, conclusions section and recommendations for future works are not include in the manuscript.

Reviewer 2 ·

Basic reporting

1. Clear and unambiguous, professional English used throughout.
- Yes. The English language of this manuscript is clear and easy to understand.

2. Literature references, sufficient field background/context provided.
- The literature references are appropriate and up-to-date.
- For field background, I suggest add a separate paragraph introducing the SVM-RFE and RF models used in this study as well as their recent applications in the section of Introduction.

3. Professional article structure, figures, tables. Raw data shared.
- This manuscript has complete structure as well as figures and tables.
- I suggest modify the title since it is a bit broad in its current form.
- I suggest add more discussions about the modeling results in Lines 234 - 239.
- I suggest add a flowchart figure to better illustrate the study.

4. Self-contained with relevant results to hypotheses.
- The results are self-contained to hypotheses.
- I suggest extend the section of Results. Only three subsections are given in the Results section while the first two of them are descriptive statistics of raw data and methods. More results regarding analysis and validation of predictions in MDD patients are needed.

Experimental design

1. Original primary research within Aims and Scope of the journal.
- Yes. This study falls within the Aims and Scope of the journal.

2. Research question well defined, relevant & meaningful. It is stated how research fills an identified knowledge gap.
- The manuscript gives well-defined questions regarding current research and identified knowledge gap.

3. Rigorous investigation performed to a high technical & ethical standard.
- Yes. This study is performed following a high technical & ethical standard.

4. Methods described with sufficient detail & information to replicate.
- I suggest give detailed information regarding the prediction process using SVM-RFE and RF models (e.g., software, parameters, etc.).

Validity of the findings

1. Impact and novelty not assessed. Meaningful replication encouraged where rationale & benefit to literature is clearly stated.
- Yes.

2. All underlying data have been provided; they are robust, statistically sound, & controlled.
- Yes.

3. Conclusions are well stated, linked to original research question & limited to supporting results.
- Yes.

Additional comments

Additional minor comments:

- Line 9: I suggest the corresponding author use official institutional email address instead of the current one, which looks a bit strange in its current form.
- Line 126: Add space before left bracket. Check and revise all.
- Line 135: Give abbreviations when first appears (e.g., RF).

---

## Round 0.2 · accepted · Accept

Thank you for addressing the reviewers' concerns.

Reviewer 1 ·

Basic reporting

- The revised manuscript is clearer and the relevant previous literature is well suited to describe the problems of the study and its contribution in comparison with other studies.
- The keywords are not provided in the revised manuscript.

Experimental design

- The sample size is clearly described using the flow chart.
- The methods described are also sufficient and can be reproduced by other researchers.

Validity of the findings

- The future works and conclusion are added in the discussion section. Not a separate section (Conclusion).